# Why So Pessimistic? Estimating Uncertainties for Offline RL through Ensembles, and Why Their Independence Matters

**Seyed Kamyar Seyed Ghasemipour**
Robotics@Google
kamyar@google.com

**Shixiang Shane Gu**
Google Research, Brain Team
shanegu@google.com

**Ofir Nachum**
Google Research, Brain Team
ofirnachum@google.com

## Abstract

Motivated by the success of ensembles for uncertainty estimation in supervised learning, we take a renewed look at how ensembles of $Q$-functions can be leveraged as the primary source of pessimism for offline reinforcement learning (RL). We begin by identifying a critical flaw in a popular algorithmic choice used by many ensemble-based RL algorithms, namely the use of shared pessimistic target values when computing each ensemble member's Bellman error. Through theoretical analyses and construction of examples in toy MDPs, we demonstrate that shared pessimistic targets can paradoxically lead to value estimates that are effectively *optimistic*. Given this result, we propose MSG, a practical offline RL algorithm that trains an ensemble of $Q$-functions with independently computed targets based on completely separate networks, and optimizes a policy with respect to the lower confidence bound of predicted action values. Our experiments on the popular D4RL and RL Unplugged offline RL benchmarks demonstrate that on challenging domains such as antmazes, MSG with deep ensembles surpasses highly well-tuned state-of-the-art methods by a wide margin. Additionally, through ablations on benchmarks domains, we verify the critical significance of using independently trained $Q$-functions, and study the role of ensemble size. Finally, as using separate networks per ensemble member can become computationally costly with larger neural network architectures, we investigate whether efficient ensemble approximations developed for supervised learning can be similarly effective, and demonstrate that they do not match the performance and robustness of MSG with separate networks, highlighting the need for new efforts into efficient uncertainty estimation directed at RL.

## 1 Introduction

Offline reinforcement learning (RL), also referred to as batch RL [1], is a problem setting in which one is provided a dataset of interactions with an environment in the form of a Markov decision process (MDP), and the goal is to learn an effective policy exclusively from this fixed dataset. Offline RL holds the promise of data-efficiency through data reuse, and improved safety due to minimizing the need for policy rollouts. As a result, offline RL has been the subject of significant renewed interest in the machine learning literature [2].

One common approach to offline RL in the model-free setting is to use approximate dynamic programming (ADP) to learn a $Q$-value function via iterative regression to backed-up *target values*. The predominant algorithmic philosophy with most success in ADP-based offline RL is to encourage

36th Conference on Neural Information Processing Systems (NeurIPS 2022).

obtained policies to remain close to the support set of the available offline data. A large variety of methods have been developed for enforcing such constraints, examples of which include regularizing policies with behavior cloning objectives [3, 4], performing updates only on actions observed inside [5, 6, 7, 8] or close to [9] the offline dataset, and regularizing value functions to underestimate the value of actions not seen in the dataset [10, 11, 12].

The need for such regularizers arises from inevitable inaccuracies in value estimation when function approximation, bootstrapping, and off-policy learning – i.e. The Deadly Triad [13] – are involved. In offline RL in particular, such inaccuracies cannot be resolved through additional interactions with the MDP. Thus, remaining close to the offline dataset limits opportunities for catastrophic inaccuracies to arise. However, recent works have argued that the aforementioned constraints can be overly pessimistic, and instead opt for approaches that take into consideration the *uncertainty* about the value function [14, 15, 16], thus re-focusing the offline RL problem to that of deriving accurate lower confidence bounds (LCB) of $Q$-values.

In the empirical supervised learning literature, *deep network ensembles* (definition in Appendix L) and their more efficient variants have been shown to be the most effective approaches for uncertainty estimation, towards learning calibrated estimates and confidence bounds with modern neural network function approximators [17]. Motivated by this, in our work we take a renewed look into $Q$-ensembles, and study how to leverage them as the primary source of pessimism for offline RL.

In deep RL, a very popular algorithmic choice is to use an ensemble of $Q$-functions to obtain pessimistic value estimates and combat overestimation bias [18]. Specifically, in the policy evaluation procedure, all $Q$-networks are updated towards a *shared pessimistic temporal difference target*. Similarly in offline RL, in addition to the main offline RL objective that they propose, several existing methods use such $Q$-ensembles [10, 3, 19, 20, 21, 22, 23, 8]. We begin by mathematically characterizing a critical flaw in the aforementioned ensembling procedure. Specifically, we demonstrate that using shared pessimistic targets can paradoxically lead to $Q$-estimates which are in fact *optimistic*! We verify our finding by constructing pedagogical toy MDPs. These results demonstrate that the formulation of using shared pessimistic targets is fundamentally ill-formed.

To resolve this problem, we propose *Model Standard-deviation Gradients* (MSG), an ensemble-based offline RL algorithm. In MSG, each $Q$-network is trained *independently*, *without sharing targets*. Crucially, ensembles trained with independent target values will always provide pessimistic value estimates. The pessimistic lower-confidence bound (LCB) value estimate – computed as the mean minus standard deviation of the $Q$-value ensemble – is then used to update the policy being trained. Evaluating MSG on the established D4RL [24] and RL Unplugged [25] benchmarks for offline RL, we demonstrate that MSG matches, and in the more challenging domains such as antmazes, significantly exceeds the prior state-of-the-art. Additionally, through a series of ablation experiments on benchmark domains, we verify the significance of our theoretical findings, study the role of ensemble size, and highlight the settings in which ensembles provide the most benefit.

The use of ensembles will inevitably be a computational bottleneck when applying offline RL to domains requiring large neural network models. Hence, as a final analysis, we investigate whether the favorable performance of MSG can be obtained through the use of modern *efficient ensemble* approaches which have been successful in the supervised learning literature [26, 27, 28, 17]. We demonstrate that while efficient ensembles are competitive with the state-of-the-art on simpler offline RL benchmark domains, similar to many popular offline RL methods they fail on more challenging tasks, and cannot recover the performance and robustness of MSG using full ensembles with separate neural networks.

Our work highlights some of the unique and often overlooked challenges of ensemble-based uncertainty estimation in offline RL. Given the strong performance of MSG, we hope our work motivates increased focus into efficient and stable ensembling techniques directed at RL, and that it highlights intriguing research questions for the community of neural network uncertainty estimation researchers whom thus far have not employed sequential domains such as offline RL as a testbed for validating modern uncertainty estimation techniques.

## 2 Related Work

Uncertainty estimation is a core component of RL, since an agent only has a limited view into the mechanics of the environment through its available experience data. Traditionally, uncertainty

estimation has been key to developing proper *exploration* strategies such as upper confidence bound (UCB) and Thompson sampling [29], in which an agent is encouraged to seek out paths where its uncertainty is high. Offline RL presents an alternative paradigm, where the agent must act conservatively and is thus encouraged to seek out paths where its uncertainty is low [14]. In either case, proper and accurate estimation of uncertainties is paramount. To this end, much research has been produced with the aim of devising provably correct uncertainty estimates [30, 31, 32], or at least bounds on uncertainty that are good enough for acting exploratorily [33] or conservatively [34]. However, these approaches require exceedingly simple environment structure, typically either a finite discrete state and action space or linear spaces with linear dynamics and rewards.

While theoretical guarantees for uncertainty estimation are more limited in practical situations with deep neural network function approximators, a number of works have been able to achieve practical success, for example using deep network analogues for count-based uncertainty [35], Bayesian uncertainty [36, 37], and bootstrapping [38, 39]. Many of these methods employ ensembles. In fact, in continuous control RL, it is common to use an ensemble of two value functions and use their minimum for computing a target value during Bellman error minimization [18]. A number of works in offline RL have extended this to propose backing up minimums or lower confidence bound estimates over larger ensembles [3, 10, 19, 20, 22, 23, 21]. In our work, we continue to find that ensembles are extremely useful for acting conservatively, but the manner in which ensembles are used is critical. Specifically our proposed MSG algorithm advocates for using independently learned ensembles, without sharing of target values, and this important design decision is supported by empirical evidence.

The widespread success of ensembles for uncertainty estimation in RL echoes similar findings in supervised deep learning. While there exist proposals for more technical approaches to uncertainty estimation [40, 41, 42], ensembles have repeatedly been found to perform best empirically [26, 43]. Much of the active literature on ensembles in supervised learning is concerned with computational efficiency, with various proposals for reducing the compute or memory footprint of training and inference on large ensembles [28, 44, 27]. While these approaches have been able to achieve impressive results in supervised learning, our empirical results suggest that their performance suffers significantly in challenging offline RL settings compared to deep ensembles.

## 3 Pessimistic $Q$-Ensembles: Independent or Shared Targets?

In this section we identify a critical flaw in how ensembles are commonly employed – in offline as well as online RL – for obtaining pessimistic value estimates [10, 3, 19, 20, 21, 22, 23, 8, 21], which can paradoxically lead to an optimism bonus! We begin by mathematically characterizing this problem and presenting a simple fix. Subsequently, we leverage our results to construct pedagogical toy MDPs demonstrating the practical importance of the identified problem and solution.

### 3.1 Mathematical Characterization

We assume access to a dataset $D$ composed of $(s, a, r, s')$ transition tuples from a Markov Decision Process (MDP) determined by a tuple $M = \langle \mathcal{S}, \mathcal{A}, \mathcal{R}, \mathcal{P}, \gamma \rangle$, corresponding to state space, action space, reward function, transitions dynamics, and discount, respectively. As is standard in RL, we do not assume any knowledge of $\mathcal{R}, \mathcal{P}$, other than that implicitly provided by the dataset $D$. In this section, for clarity of exposition, we assume that the policies we consider are *deterministic*, and that our MDPs do not have terminal states.

We consider $Q$-value ensemble members given by a parameterization $Q_{\theta^i}$, where $i$

1. Initialize $\theta^i$ for all $i \in Z$.

2. For $t = 1, 2, \dots$:
   - For each $(s, a, r, s') \in D$ and $i \in Z$ compute target values $y^i(r, s', \pi)$.
   - For each $i \in Z$, update $\theta^i$ to optimize the regression objective

     $$\frac{1}{|D|} \sum_{(s,a,r,s') \in D} (Q_{\theta^i}(s,a) - y^i(r, s', \pi))^2$$

3. Return a pessimistic $Q$-value function $Q_{\text{pessimistic}}$ based on the trained ensemble.

indexes into some set $Z$, which is finite in practice but may be infinite or uncountable in theory. We assume $Z$ has an associated probability space allowing us to make expectation $\mathbb{E}$ or variance $\mathbb{V}$

computations over the ensemble members. Given a fixed policy $\pi$, a general dynamic programming based procedure for obtaining pessimistic value estimates is outlined by the iterative regression described in the box above.

A key algorithmic choice in this recipe is where pessimism should be introduced. This can be done by either (a) pessimistically aggregating $Q$-values after training, i.e. inside Step 3, or (b) *also incorporating pessimism during Step 2,* by using a shared pessimistic target value $y$. Through our review of the offline RL (as well as online RL) literature, we have observed that the most common approach is the latter, where the targets are *pessimistic, shared, and identical* across ensemble members [10, 3, 19, 20, 21, 22, 23, 8]. Specifically, they are computed as, $y^i(r, s', \pi) = \text{PO}(\{r + \gamma Q_{\theta^i}(s', \pi(s')), \forall i \in Z\})$ with PO being a desired pessimism operator aggregating the TD target values of the ensemble members (e.g. "mean minus standard deviation", or "minimum").

In this section, our goal is to compare these two alternative approaches. For our analysis, we will use "mean minus standard deviation" (a lower confidence bound (LCB)) as our pessimism operator, and use the notation $Q_{\text{LCB}}$ in place of $Q_{\text{pessimistic}}$ (defined in the box above). Under the LCB pessimism operator we will have:

**Independent Targets (Method 1):** $\quad y^i(r, s', \pi) = r + \gamma \cdot Q_{\theta^i}(s', \pi(s'))$

**Shared Targets (Method 2):** $\quad y^i(r, s', \pi) = r + \gamma \cdot \left( \mathbb{E}_{\text{ens}}\left[ Q_{\theta^i}(s', \pi(s')) \right] - \sqrt{\mathbb{V}_{\text{ens}}\left[ Q_{\theta^i}(s', \pi(s')) \right]} \right)$

**For both we have:** $\quad Q_{\text{LCB}}(s, a) = \mathbb{E}_{\text{ens}}\left[ Q_{\theta^i}(s, a) \right] - \sqrt{\mathbb{V}_{\text{ens}}\left[ Q_{\theta^i}(s, a) \right]}$

To characterize the form of $Q_{\text{LCB}}$ when using complex neural networks, we refer to the work on infinite-width neural networks, namely the Neural Tangent Kernel (NTK) [45]. We consider $Q$-value ensemble members, $Q_{\theta^i}$, which all share the same infinite-width neural network architecture (and thus the same NTK parameterization). As noted in the algorithm box above, and as is the case in deep ensembles [43], the only difference amongst ensemble members $Q_{\theta^i}$ is in their initial weights $\theta^i$ sampled from the neural network's initial weight distribution.

Before presenting our results, we establish some notation relevant to the infinite-width and NTK regime. Let $\mathcal{X}, R, \mathcal{X}'$ denote data matrices containing $(s, a)$, $r$, and $(s', \pi(s'))$ appearing in the offline dataset $D$; i.e., the $k$-th transition $(s, a, r, s')$ in $D$ is represented by the $k$-th rows in $\mathcal{X}, R, \mathcal{X}'$. Let $A, B$ denote two data matrices, where similar to $\mathcal{X}, \mathcal{X}'$, each row contains a state-action tuple $(s, a) \in \mathcal{S} \times \mathcal{A}$. The NTK, which governs the training dynamics of the infinitely-wide neural network, is then given by the outer product of gradients of the neural network *at initialization*: $\hat{\Theta}_i^{(0)}(A, B) := \nabla_\theta Q_{\theta^i}(A) \cdot \nabla_\theta Q_{\theta^i}(B)^T|_{t=0}$, where we overload notation $Q_{\theta^i}(A)$ to represent the column vector containing $Q$-values. At infinite-width in the NTK regime, $\hat{\Theta}_i^{(0)}(A, B)$ converges to a deterministic kernel (i.e. does not depend on the random weight sample $\theta^i$), and hence is the same for all ensemble members. Thus, hereafter we will remove the index $i$ from the notation of the NTK kernel and simply write, $\hat{\Theta}^{(0)}(A, B)$. With our notation in place, we define, $C := \hat{\Theta}^{(0)}(\mathcal{X}', \mathcal{X}) \cdot \hat{\Theta}^{(0)}(\mathcal{X}, \mathcal{X})^{-1}$. Intuitively, $C$ is a $|D| \times |D|$ matrix where the element at column $q$, row $p$, captures a notion of similarity between $(s, a)$ in the $q^{th}$ row of $\mathcal{X}$, and $(s', \pi(s'))$ in the $p^{th}$ row of $\mathcal{X}'$.

We now have all the necessary machinery to characterize the form of $Q_{\text{LCB}}$:

**Theorem 3.1.** *For a given $(s, a) \in \mathcal{S} \times \mathcal{A}$, let $Q_{\theta^i}^{(0)}(s, a)$ denote $Q_{\theta^i}(s, a)|_{t=0}$ (value at initialization), with $\theta$ sampled from the initial weight distribution. After $t + 1$ iterations of pessimistic policy evaluation, the LCB value estimate for $(s', \pi(s')) \in \mathcal{X}'$ is given by,*

***Independent Targets (Method 1):*** $\hfill (1)$

$$Q_{\text{LCB}}^{(t+1)}(\mathcal{X}') = \mathcal{O}(\gamma^t \|C\|^t) + \underbrace{(1 + \ldots + \gamma^t C^t)}_{backup\ term} CR - \sqrt{\mathbb{E}_{\text{ens}}\left[ \left( \underbrace{(1 + \ldots + \gamma^t C^t)}_{backup\ term}(Q_{\theta^i}^{(0)}(\mathcal{X}') - CQ_{\theta^i}^{(0)}(\mathcal{X})) \right)^2 \right]}$$

***Shared Targets (Method 2):*** $\hfill (2)$

$$Q_{\text{LCB}}^{(t+1)}(\mathcal{X}') = \mathcal{O}(\gamma^t \|C\|^t) + \underbrace{(1 + \ldots + \gamma^t C^t)}_{backup\ term} CR - \underbrace{(1 + \ldots + \gamma^t C^t)}_{backup\ term} \sqrt{\mathbb{E}_{\text{ens}}\left[ \left( Q_{\theta^i}^{(0)}(\mathcal{X}') - CQ_{\theta^i}^{(0)}(\mathcal{X}) \right)^2 \right]}$$

*where the square and square-root operations are applied element-wise.*[1] *Please refer to Appendix F for the proof.*

As can be seen, the equations for the pessimistic LCB value estimates in both settings are similar, only differing in the third term. The first term is negligible and tends towards zero as the number of iterations of policy evaluation increases. The second term shared by both variants corresponds to the expected result of the policy evaluation procedure *without any pessimism* (as before, we mean expectation under $\theta$ sampled from the initial weight distribution). Accordingly, the differing third term in each variant exactly corresponds to the "pessimism" or "penalty" induced by that variant.

Considering the available offline RL dataset $D$ as a restricted MDP in itself, we see that the use of Independent Targets (Method 1) leads to a pessimism term that performs "backups" along the trajectories that the policy would experience in this restricted MDP (using the geometric term $1 + \cdots + \gamma^t C^t$) before computing a variance estimate. Meanwhile the use of Shared Targets (Method 2) does the reverse – it first computes a variance term and then performs the "backups".

While this difference may seem inconsequential, it becomes critical when one realizes that in Equation 2 for Shared Targets (Method 2), the pessimism term (third term) may become positive, i.e. a *negative penalty*, yielding an effectively *optimistic* LCB estimate. Critically, with Independent Targets (Method 1), this problem *cannot occur*.

### 3.2 Validating Theoretical Predictions

In this section we demonstrate that our analysis is not solely a theoretical result concerning the idiosyncracies of infinite-width neural networks, but that it is rather straightforward to construct combinations of an MDP, offline data, and a policy, that lead to the critical flaw of an optimistic LCB estimate.

Let $d_s, d_a$ denote the dimensionality of state and action vectors respectively. We consider an MDP whose initial state distribution is a spherical multivariate normal distribution $\mathcal{N}(0, I)$, and whose transition function is given by $\mathcal{P}(s'|s, a) = \mathcal{N}(0, I)$. Consider the procedure for generating our offline data matrices, described in the box to the right. This procedure

1. Initialize empty $\mathcal{X}, R, \mathcal{X}'$
2. For $N$ episodes:
   - sample $s \sim \mathcal{N}(0, I)$
   - For $T$ steps:
     - sample $a \sim \mathcal{N}(0, I)$
     - sample $s' \sim \mathcal{N}(0, I)$
     - set $\pi(s') \leftarrow a$
     - Add $(s, a)$ to $\mathcal{X}$
     - Add $r \sim \mathcal{N}(0, I)$ to $R$
     - Add $(s', \pi(s'))$ to $\mathcal{X}'$
     - Set $s \leftarrow s'$
3. Return the offline dataset $\mathcal{X}, R, \mathcal{X}'$

returns data matrices $\mathcal{X}, R, \mathcal{X}'$ by generating $N$ episodes of length $T$, using a behavior policy $a \sim \mathcal{N}(0, I)$. In this generation process, we set the policy we seek to pessimistically evaluate, $\pi$, to always apply the behavior policy's action in state $s$ to the next state $s'$.

To construct our examples, we consider the setting where we use linear models to represent $Q_{\theta^i}$, with the initial weight distribution being a spherical multivariate normal distribution, $\mathcal{N}(0, I)$. With linear models, the equations for $Q_{\text{LCB}}$ takes an identical form to those in Theorem 3.1.

Given the described data generating process and our choice of linear function approximation, we can compute the pessimism term for the Shared Targets (Method 2) (i.e. the third term in Theorem 3.1, Equation 2). We implement this computation in a simple Python script, which we include in the supplementary material. We choose, $d_s = 30$, $d_a = 30$, $\gamma = 0.5$, $N = 5$, $T = 5$, and $t = 1000$ ($t$ is the exponent in the geometric term above). We run this simulation 1000 times, each with a different random seed. After filtering simulation runs to ensure $\gamma\|C\| < 1$ (as discussed in an earlier footnote), we observe that 221 of the simulation runs result in an optimistic LCB bonus, meaning that in those experiments, the pessimism term was in fact positive for some $(s', \pi(s')) \in \mathcal{X}'$. We have made the python notebook implementing this experiment available in our supplementary material. For further intriguing investigations in pedagogical toy MDPs regarding the structure of uncertainties, we strongly encourage the interested reader to refer to Appendix G.

---

[1]Note that if $\gamma\|C\| \geq 1$, dynamic programming is liable to diverge in either setting. In our discussions, we avoid this degenerate case and assume $\gamma\|C\| < 1$.

# 4 Model Standard-deviation Gradients (MSG)

It is important to note that even if the pessimism term does not become positive for a particular combination of MDPs, offline datasets, and policies, the fact that it can occur highlights that the formulation of Shared Targets is fundamentally ill-formed. To resolve this problem we propose Model Standard-deviation Gradients (MSG), an offline RL algorithm which leverages ensembles to approximate the LCB using the approach of Independent Targets.

## 4.1 Policy Evaluation and Optimization in MSG

MSG follows an actor-critic setup. At the beginning of training, we create an ensemble of $N$ Q-functions by taking $N$ samples from the initial weight distribution. During training, in each iteration, we first perform policy evaluation by estimating the $Q_{\text{LCB}}$ for the current policy, and subsequently optimize the policy through gradient ascent on $Q_{\text{LCB}}$.

**Policy Evaluation**    As motivated by our analysis in Section 3, we train the ensemble $Q$-functions independently using the standard least-squares Q-evaluation loss,

$$\mathcal{L}(\theta^i) = \mathbb{E}_{(s,a,r,s')\sim D}\Big[ \big(Q_{\theta^i}(s,a) - y^i(r,s',\pi)\big)^2\Big]; \quad y^i = r + \gamma \cdot \mathbb{E}_{a'\sim\pi(s')}\Big[Q_{\bar{\theta}^i}(s',a')\Big] \quad (3)$$

where $\theta^i, \bar{\theta}^i$ denote the parameters and target network parameters for the $i^{\text{th}}$ Q-function.

In each iteration, as is common practice, we do not update the $Q$-functions until convergence, and instead update the networks using a single gradient step. In practice, the expectation in $\mathcal{L}(\theta^i)$ is estimated by a minibatch, and the expectation in $y^i$ is estimated with a single action sample from the policy. After every update to the Q-function parameters, their corresponding target parameters are updated to be an exponential moving average of the parameters in the standard fashion.

**Policy Optimization**    As in standard deep actor-critic algorithms, policy evaluation steps (learning $Q$) are interleaved with policy optimization steps (learning $\pi$). In MSG, we optimize the policy through gradient ascent on $Q_{\text{LCB}}$. Specifically, our proposed policy optimization objective in MSG is,

$$\mathcal{L}(\pi) = \mathbb{E}_{s\sim D, a\sim\pi(s)}\left[Q_{\text{LCB}}(s,a)\right] = \mathbb{E}_{s\sim D, a\sim\pi(s)}\left[\mathbb{E}_{\text{ens}}[Q_{\theta^i}(s,a)] + \beta\sqrt{\mathbb{V}_{\text{ens}}[Q_{\theta^i}(s,a)]}\right] \quad (4)$$

where $\beta \leq 0$ is a hyperparameter that determines the amount of pessimism.

## 4.2 The Trade-Off Between Trust and Pessimism

While our hope is to leverage the implicit generalization capabilities of neural networks to estimate proper LCBs beyond states and actions in the finite dataset $D$, neural network architectures can be fundamentally biased, or we can simply be in a setting with insufficient data coverage, such that the generalization capability of those networks is limited. To this end, we augment the policy evaluation objective of MSG ($\mathcal{L}(\theta^i)$, equation 3) with a support constraint regularizer inspired by CQL [11] [2]: $\mathcal{H}(\theta^i) = \mathbb{E}_{s\sim D, a\sim\pi(s)}\left[Q_{\theta^i}(s,a)\right] - \mathbb{E}_{(s,a)\sim D}\left[Q_{\theta^i}(s,a)\right]$. This regularizer encourages the $Q$-functions to increase the values for actions seen in the dataset $D$, while decreasing the values of the actions of the current policy. Practically, we estimate the latter expectation of $\mathcal{H}$ using the states in the mini-batch, and we approximate the former expectation using a single sample from the policy. We control the contribution of $\mathcal{H}(\theta^i)$ by weighting this term with weight parameter $\alpha$. The full critic loss is thus given by,

$$\mathcal{L}(\theta^1, \ldots, \theta^N) = \sum_{i=1}^{N} \big(\mathcal{L}(\theta^i) + \alpha\mathcal{H}(\theta^i)\big) \quad (5)$$

Empirically, as evidenced by our results in Appendix A.2, we have observed that such a regularizer can be necessary in two situations: 1) The first scenario is where the offline dataset only contains a narrow data distribution (e.g., imitation learning datasets only containing expert data). We believe

---

[2] Instead of a CQL-style value regularizer, other forms of support constraints such as a behavioral cloning regularizer on the policy could potentially be used.

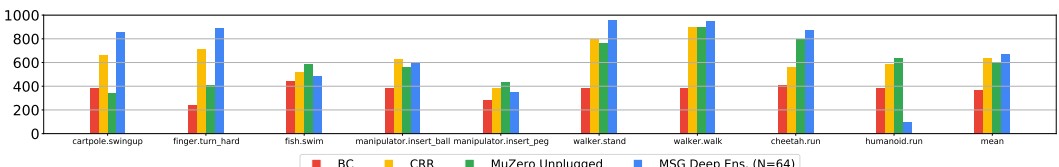

Figure 1: Results for DM Control Suite subset of the RL Unplugged benchmark [25]. We note that: 1) the architecture used for MSG is smaller by a factor of approximately 60x which contributes to poor performance on `humanoid.run`, 2) CRR results are reported by their best checkpoint throughout training which differs from MSG, BC, and MuZero Unplugged which report performance at the end of training. Baseline results taken from [47]. Despite MSG's disadvantage on the `humanoid.run` task, it still edges out the baseline methods in mean performance.

| Domain | CQL | IQL | MSG ($N=64$) | $\beta$ | $\alpha$ | MSG ($N=4$) | $\beta$ | $\alpha$ |
|---|---|---|---|---|---|---|---|---|
| maze2d-umaze-v1 | 5.7 | – | **$101.1 \pm 26.3$** | $-8$ | $0$ | $68.8 \pm 20.2$ | $-4$ | $0.1$ |
| maze2d-medium-v1 | 5.0 | – | **$57.0 \pm 17.2$** | $-4$ | $0.1$ | **$53.2 \pm 26.8$** | $-4$ | $0.1$ |
| maze2d-large-v1 | 12.5 | – | **$159.3 \pm 49.4$** | $-4$ | $0.1$ | $59.2 \pm 59.1$ | $-8$ | $0.5$ |
| antmaze-umaze-v0 | 74.0 | 87.5 | **$97.8 \pm 1.2$** | $-4$ | $0.5$ | **$98.6 \pm 1.4$** | $-4$ | $1.0$ |
| antmaze-umaze-diverse-v0 | **84.0** | 62.2 | $81.8 \pm 3.0$ | $-4$ | $1.0$ | $76.6 \pm 7.6$ | $-4$ | $0.5$ |
| antmaze-medium-play-v0 | 61.2 | 71.2 | **$89.6 \pm 2.2$** | $-4$ | $0.5$ | **$83.0 \pm 7.1$** | $-4$ | $0.1$ |
| antmaze-medium-diverse-v0 | 53.7 | 70.0 | **$88.6 \pm 2.6$** | $-4$ | $0.5$ | **$83.0 \pm 6.2$** | $-4$ | $0.5$ |
| antmaze-large-play-v0 | 15.8 | 39.6 | **$72.6 \pm 7.0$** | $-8$ | $0$ | $46.8 \pm 14.7$ | $-4$ | $0.5$ |
| antmaze-large-diverse-v0 | 14.9 | 47.5 | **$71.4 \pm 12.2$** | $-8$ | $0.1$ | $58.2 \pm 9.6$ | $-8$ | $0.1$ |

Table 1: Results on D4RL maze2d and antmaze domains. In MSG, $\beta$ is the hyperparameter controlling the amount of pessimism in $Q_{LCB}$ (Equation 4), and $\alpha$ is the hyperparameter controlling the contribution of the CQL-style regularizer (Equation 5). As we were unable to reproduce CQL antmaze results despite extensive hyperparameter tuning (see also [12]), we present the numbers reported by the original paper which uses the same network architectures as MSG. We also present reported results for the current state-of-the-art, IQL [48].

this is because the power of ensembles comes from predicting a value distribution for unseen $(s, a)$ based on the available training data. Thus, if no data for sub-optimal actions is present, ensembles cannot make accurate predictions and increased pessimism via $\mathcal{H}$ becomes necessary. 2) The second scenario is where environment dynamics can be chaotic (e.g. Gym [46] `hopper` and `walker2d`). In such domains it would be beneficial to remain close to the observed data in the offline dataset. Pseudo-code for our proposed MSG algorithm can be viewed in Algorithm Box 1.

## 5 Experiments

In this section we seek to empirically answer the following questions: 1) How well does MSG perform compared to current state-of-the-art in offline RL? 2) Are the theoretical differences in ensembling approaches (Section 3) practically relevant? 3) When and how does ensemble size affect perfomance? 4) Can we match the performance of MSG through efficient ensemble approximations developed in the supervised learning literature?

### 5.1 Offline RL Benchmarks

**D4RL Gym Domains** We begin by evaluating MSG on the Gym domains (`halfcheetah`, `hopper`, `walker2d`) of the D4RL offline RL benchmark [24], using the `medium`, `medium-replay`, `medium-expert`, and `expert` data settings. Our results presented in Appendix A.2 (summarized in Figure 4) demonstrates that MSG is competitive with well-tuned state-of-the-art methods CQL [11] and F-BRC [12].

**D4RL Antmaze Domains** Due to the narrow range of behaviors in Gym environments, offline datasets for these domains tend to be very similar to imitation learning datasets. As a result, many prior offline RL approaches that perform well on D4RL Gym fail on harder tasks that require stitching trajectories through dynamic programming (c.f. [48]). An example of such tasks are the D4RL antmaze settings, in particular those in the `antmaze-medium` and `antmaze-large` environments. The data for antmaze tasks consists of many episodes of an Ant agent [46] running along arbitrary

paths in a maze. The agent is tasked with using this data to learn a point-to-point navigation policy from one corner of the maze to the opposite corner, where rewards are given by a sparse signal that is 1 when near the desired end location in the maze – at which point the episode is terminated – and 0 otherwise. The undirected, extremely sparse reward nature of antmaze tasks make them very challenging, especially for the large maze sizes.

Table 1 and Appendix B.2 present our results. To the best of our knowledge, the antmaze domains are considered unsolved, with few prior works reporting non-zero results on the large mazes [11, 48]. As can be seen, MSG obtains results that far exceed the prior state-of-the-art results reported by [48]. While some works that use specialized hierarchical approaches have reported strong results as well [49], it is notable that MSG is able to solve these challenging tasks with standard architectures and training procedures, and this shows the power that ensembling can provide – as long as the ensembling is performed properly!

**RL Unplugged**  In addition to the D4RL benchmark, we evaluate MSG on the RL Unplugged benchmark [25]. Our results are presented in Figure 1. We compare to results for Behavioral Cloning (BC) and two state-of-the-art methods in these domains, Critic-Regularized Regression (CRR) [7] and MuZero Unplugged [47]. Due to computational constraints when using deep ensembles, we use the same network architectures as we used for D4RL experiments. The networks we use are approximately $\frac{1}{60}$-th the size of those used by the BC, CRR, and MuZero Unplugged baselines in terms of number of parameters. We observe that MSG is on par with or exceeds the current state-of-the-art on all tasks with the exception of `humanoid.run`, which appears to require the larger architectures used by the baseline methods. Experimental details can be found in Appendix C.

**Benchmark Conclusion**  Prior work has demonstrated that many offline RL approaches that perform well on Gym domains, fail to succeed on much more challenging domains [48]. Our results demonstrate that through uncertainty estimation with deep ensembles, MSG is able to very significantly outperform prior work on very challenging benchmark domains such as the D4RL antmazes.

## 5.2  Ensemble Ablations

**Independence in Ensembles Ablation**  In Section 3, through theoretical arguments and toy experiments we demonstrated the importance of training using "Independent" ensembles. Here, we seek to validate the significance of our theoretical findings using offline RL benchmarks, by comparing `Independent` targets (as in MSG), to `Shared-LCB` and `Shared-Min` targets. Our results are presented in Appendices A.3 and B.3, with a summary in Figures 3 and 4.

In the Gym domains (Appendix A.3), with ensemble size $N = 4$, `Shared-LCB` significantly underperforms MSG. In fact, not using ensembles at all ($N = 1$) outperforms `Shared-LCB`. With ensemble size $N = 4$, `Shared-Min` is on par with MSG. When the ensemble size is increased to $N = 64$ (Figure 7), we observe the performance of `Shared-Min` drops significantly on $7/12$ D4RL Gym settings. In constrast, the performance of MSG is stable and does not change.

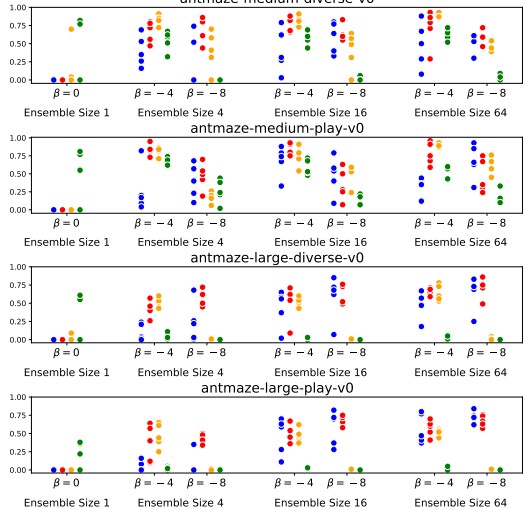

Figure 2:  Ensemble size ablation on the antmaze medium and large domains with varying $\beta \in \{-4, -8\}$ and $\alpha \in \{0, 0.1, 0.5, 1\}$ (colored blue, red, yellow, and green respectively). We observe the general trend that bigger ensembles lead to better performance.

In the challenging antmaze domains (Appendix B.3), for both ensemble sizes $N = 4$ and $N = 64$, `Shared-LCB` and `Shared-Min` targets completely fail to solve the tasks, while for both ensemble sizes MSG exceeds the prior state-of-the-art (Table 1), IQL [48].

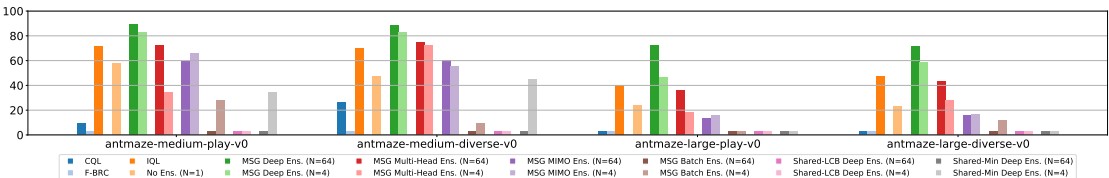

Figure 3: Summary of D4RL antmaze benchmark results (full results presented in Appendix B). For each method, we report the mean across random seeds for the best hyperparameter. Numerical results for all experiments are available in the supplementary material.

**Independence in Ensembles Conclusion** Our experiments corroborate the theoretical results in Section 3, demonstrating that `Independent` targets are critical to the success of MSG. These results are particularly striking when one considers that the implementations for MSG, `Shared-LCB`, and `Shared-Min` differ by only 2 lines of code.

**Ensemble Size Ablation** An important ablation is to understand the role of ensemble size in MSG.

In the Gym domains, Figure 5 demonstrates that increasing the number of ensembles from $4$ to $64$ does not result in a noticeable change in performance.

In the antmaze domains, we evaluate MSG under ensemble sizes $\{1, 4, 16, 64\}$. Figure 2 presents our results. Our key takeaways are as follows:

- For the harder `antmaze-large` tasks, there is a clear upward trend as ensemble size increases.
- Using a small ensemble size (e.g. $N = 4$) is already quite good, but more sensitive to hyperparameter choice especially on the harder tasks.
- Very small ensemble sizes benefit more from using $\alpha > 0$ [3]. However, across the board, using $\alpha = 0$ is preferable to using too large of a value for $\alpha$ – with the exception of $N = 1$ which cannot take advantange of the benefits of ensembling.
- When using lower values of $\beta$, lower values of $\alpha$ should be used.

**Ensemble Size Conclusion** In domains such as D4RL Gym where offline datasets are qualitatively similar to imitation learning datasets, larger ensembles do not result in noticeable gains. In domains such as D4RL antmaze which contain more data diversity, larger ensembles significantly improve the performance of agents.

### 5.3 Efficient Ensembles

Thus far we have demonstrated the significant performance gains attainable through MSG. An important concern however, is that of parameter and computational efficiency: Deep ensembles of $Q$-networks result in an $N$-fold increase in memory and compute usage, both in the policy evaluation and policy optimization phases of actor-critic training. While this might not be a significant problem in offline RL benchmark domains due to small model footprints[4], it becomes a major bottleneck with larger architectures such as those used in language and vision domains. To this end, we evaluate whether recent advances in "Efficient Ensemble" approaches from the supervised learning literature transfer well to the problem of offline RL. Specifically, the efficient ensemble approaches we consider are: Multi-Head Ensembles [26, 50, 51], MIMO Ensembles [27], and Batch Ensembles [28]. For a description of these efficient ensembling approaches please refer to Appendix E. A runtime comparison of different ensembling approaches can be viewed in Table 2.

**D4RL Gym Domains** Appendix A.4 presents our results in the D4RL Gym domains with ensemble size $N = 4$ (summary in Figure 4). Amongst the considered efficient ensemble approaches, Batch Ensembles [28] result in the best performance, which follows findings from the supervised learning literature [17].

---

[3]As a reminder, $\alpha$ is the weight of the CQL-style regularizer loss discussed in Section 4.2.

[4]All our experiments were ran on a single Nvidia P100 GPU.

**D4RL Antmaze Domains**    Appendix B.4 presents our results in the D4RL antmaze domains for both ensemble sizes of $N = 4$ and $N = 64$ (summary in Figure 3). As can be seen, compared to MSG with deep ensembles (separate networks), the efficient ensemble approaches we consider are very unreliable, and fail for most hyperparameter choices.

**Efficient Ensembles Conclusion**    We believe the observations in this section very clearly motivate future work in developing efficient uncertainty estimation approaches that are better suited to the domain of reinforcement learning. To facilitate this direction of research, in our codebase we have included a complete boilerplate example of an offline RL agent, amenable to drop-in implementation of novel uncertainty-estimation techniques.

## 6    Discussion & Future Work

Our work has highlighted the significant power of ensembling as a mechanism for uncertainty estimation for offline RL. In this work we took a renewed look into $Q$-ensembles, and studied how to leverage them as the primary source of pessimism for offline RL. Through theoretical analyses and toy constructions, we demonstrated a critical flaw in the popular approach of using shared targets for obtaining pessimistic $Q$-values, and demonstrated that it can in fact lead to optimistic estimates. Using a simple fix, we developed a practical deep offline RL algorithm, MSG, which resulted in large performance gains on established offline RL benchmarks.

As demonstrated by our experimental results, an important outstanding direction is to study how we can design improved efficient ensemble approximations, as we have demonstrated that current approaches used in supervised learning are not nearly as effective as MSG with ensembles that use separate networks. We hope that this work engenders new efforts from the community of neural network uncertainty estimation researchers towards developing efficient uncertainty estimation techniques directed at reinforcement learning.

## Acknowledgments and Disclosure of Funding

We would like to thank Yasaman Bahri for insightful discussions regarding infinite-width neural networks. We would like to thank Laura Graesser for providing a detailed review of our work. We would like to thank conference reviewers for posing important questions that helped clarify the organization of this manuscript.

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
