# OpenReview forum: "Why So Pessimistic? Estimating Uncertainties for Offline RL through Ensembles, and Why Their Independence Matters"
_NeurIPS.cc/2022/Conference — NeurIPS 2022 Accept_

### Official Review · Reviewer_nkxP · 2022-07-07

**Rating:** 8
**Confidence:** 4
**Soundness:** 4 excellent
**Presentation:** 4 excellent
**Contribution:** 3 good

**Summary:**

The paper observes a problem in existing pessimism estimation in offline RL using ensembles: using shared targets for all ensembles updates. The paper instead proposes to update each ensemble individually and apply the pessimism at policy updates. The paper derives the update form of both methods in the NTK setting and shows that the update method with shared target could even result in optimism, which is also shown with some synthetic simulation data. Finally the paper evaluates the proposed method in several offline RL benchmarks and show its empirical competitiveness.

**Questions:**

## Major Questions

1. This question is less about the proposed algorithm itself: for the shared target update methods, the target itself contains a pessimism term: it's already a LCB estimation. Why, for policy update, another LCB term is added into the $Q_{LCB}$ estimation, why not just directly using $\mathbb{E}[Q_{\theta^i}]$? It seems like it's doing pessimism twice? Although it looks like removing the LCB term in $Q_{LCB}$ won't affect the theoretical result by much (you just subtract the backup term by 1? which of course leads to optimism more easily).

2. In addition to the existing baselines, it may seem like the following baselines could also be of interest? For example, a) we take the min over the ensemble with individual targets? b) every ensemble is bootstrapped from their own target but we also subtract with a common lcb term?

### some minor problems

- The formula in the algorithm box in section 3.1 overflows.

- Why the $\gamma$ in section 3.2 is chosen to be very small (instead of 0.9)?

- The y-axis are inconsistent in Fig.2 and Fig.3. Also in Fig.3, what does each dot mean? It seems a little bit hard to interpret the plot.

- Why is line 763 true, for $\theta_{lin}^{(0)} = 0$?

**Limitations:**

1. The overall algorithm has good intuition and motivation, but the introduction to the additional term in section 4.2 looks irrelevant to the rest of the paper. From the experiments, this term seems crucial to a good performance of the algorithm and thus unavoidable in the current version. Although it makes sense that some kind of regularization may be needed for unseen action, this additional term indeed undermines the overall message a little bit.

2. The ablation of using a more condensed surrogate for ensemble is a good experiment, and as the paper already suggests, it would be better if a more efficient way of pessimism could be derived, which seems beyond the scope of this paper.

**Strengths And Weaknesses:**

## Strength

1. Overall the paper is well written, easy to follow, and the technical part seems correct.

2. The paper makes a good observation of the existing methods for offline RL when they update the Q-values for ensembles: from hindsight one really does not need to incorporate the pessimism into the function update procedure, but instead just apply pessimism during policy update. This also seems to agree to the theory RL algorithms: one can just perform the regular bellman updates (or perform elimination in version space algorithms) and define policy with LCB or take minimum over the remaining set of functions (for pessimism) (for example, [1]).

3. Although it may not be obvious under what kind of conditions such that in the NTK setting, using the shared target could result in optimism, the following subsection provides good evidence that that indeed could happen. It could be better to provide some more intuitive scenario or even a closed-form construction.

4. The paper provides extensive and convincing experiments, including a) good ablation experiments which contains the different kinds of shared target updating methods (such as shared-LCB Ens., Shared-Min Deep Ens, and with a different number of ensembles)
. b) The paper tries many different hyperparameters for the baselines, so the baselines seem to be fine-tuned for the final presentation of the results. c) The experiments are performed on extensive benchmarks.

## Weakness

1. The theoretical results provide very good intuition into the problems of the previous pessimism estimation in offline deep RL methods, but since the result is based on the NTK setting, it still has some gap between the practical situations.

2. The result presented in table 1 has different hyperparameter for different tasks, which likely undermines the empirical merits of the proposed algorithm.

### references

[1] Xie, Tengyang, et al. "Bellman-consistent pessimism for offline reinforcement learning." Advances in neural information processing systems 34 (2021): 6683-6694.

---

> ### Author Response · Authors · 2022-08-04
> **Response to Reviewer nkxP (Part 1)**
>
> Thank you for your detailed review of our work! We hope that our response below addresses your concerns. We look forward to a productive Author-Reviewer discussion period.
>
> - Response to Strengths:
>   - More intuitive toy experiments for demonstrating the optimism problem of Shared-Targets:
>     - Unfortunately, whether overestimation happens or not is due to a very complex interaction between the C matrix, the neural network architecture, the initial weight distribution used to initialize the weights of the networks in the ensemble (since it affects the distributions of $Q^{(0)}(s,a)$), the data available in the offline dataset, and the policy being evaluated. The C matrix is itself a direct function of the offline dataset + the neural network architecture + the policy being evaluated: $C := \hat{\Theta}^{(0)}(X’, X) \cdot \hat{\Theta}^{(0)}(X, X)$, where $X’ := \text{data matrix }(s’, \pi(s’))$, $X := \text{data matrix }(s,a)$, and $\hat{\Theta}$ is the NTK kernel corresponding to the specific neural network architecture used. Due to such complex interactions, it can be quite challenging to intuitively understand when specifically the problem with optimism can occur. Looking at the equations, if the C matrix only contains non-negative entries then it would avoid the optimism problem with Shared-Targets (however, it will not resolve the problem of “correctness” of Shared-Targets, in the sense that they do not correspond to a notion of LCB on a pseudo-posterior of Q-functions). A special case would be if the C matrix was a diagonal matrix with non-negative entries. A special case where this would happen is if, for all $(s,a)$, $(s,a)$ is only strongly correlated with itself and $(s’, \pi(s’)$, with the notion of correlation coming from the NTK, i.e. $\hat{\Theta}^{(0)}(X, X)$ and $\hat{\Theta}^{(0)}(X’, X)$.
>     - Appendix G contains another set of toy experiments that we believe provide very interesting intuitions regarding shared vs. independent targets (unfortunately we were not able to include them in the main text). However, they are not studying the optimism problem discussed in Section 3.2.
> - Response to Weaknesses:
>   - 1) Regarding the relevance of NTK results for more practical settings: We acknowledge that the idealized NTK setting has important gaps with the practical deep RL setting, such as Adam optimizer vs. SGD, the use of target Q-functions, finite sized neural networks, few gradient steps vs. optimization till convergence in the policy evaluation step of each iteration, etc. Additionally, experiments in Appendix G.2 reveal intriguing but unexplained behaviors when moving from finite-width towards infinite-width networks. Nonetheless, NTK appears to be one of the best tools currently available for taming the complexity of neural networks, and theoretical considerations from this setting have resulted in improvements in practical deep RL settings [1]. We believe our extensive experimental results on popular offline RL benchmarks also exemplify this transfer.

---

> > ### Author Response · Authors · 2022-08-04
> > **Response to Reviewer nkxP (Part 2)**
> >
> > - Response to Weaknesses:
> >   - 2) Regarding results of Table 1:
> >     - In Table 1, for each domain we reported the results for the hyperparameter choice with the best mean performance (mean across experiments with 5 random seeds) (as a reminder, for all baselines we reported either the best hyperparameter we could find with equal hyperparameter tuning budget, or the results reported by the original authors if they were better than the results we obtained ourselves).
> >     - Figure 3 provides the broader picture. Figure 3 presents the results for 5 random seeds, per ($\alpha$, $\beta$, ensemble size) hyperparameter combination, with each dot representing one experiment. In Figure 3, the following can be clearly observed. 1) As the ensemble size increases, results improve, in particular for the two hardest antmaze-large domains, showing the significant impact of offline RL through ensembles, as larger ensembles are closer to the theoretical setting. 2) Looking at ensemble size 64, for the two antmaze-large domains almost all of the blue dots ($\alpha=0$, no CQL-style regularizer) are obtaining SOTA results, and for the two antmaze-medium domains, we see that by increasing the amount of pessimism only through the LCB (i.e. setting $\beta=-8$), the blue dots ($\alpha=0$, no CQL-style regularizer) are again amongst the best results we obtain. This again, supports the statement that uncertainty estimation through ensembles can indeed be used as the main, and only, source of pessimism for offline RL. 3) The ablation results in Figure 13 (in the appendix) demonstrate that if we modify the implementation of ensembles in 1 line of code, and go from Independent Ensembles to Shared-LCB or Shared-Min Ensembles, the SOTA results we obtained become 0 across the board, for every domain, hyperparameter, and random seed. This ablation undeniably demonstrates the fundamental practical significance of Theorem 3.1 and Section 3.2.
> >   - The main domains where we have observed the need for $\alpha > 0$ is in the D4RL gym domains (HalfCheetah, Hopper, Walker2d). In prior work such as EMaQ [2], it has been shown that performant policies for these domains can be surprisingly close to a behavioral cloning policy. Additionally, in IQL [3], it has been shown that prior methods such as 10% BC, Decision Transformers [4], AWAC [5], Onestep RL [6], TD3+BC [7], which are very close to behavioral cloning, do very well on these gym domains, while completely failing on on the antmaze domains. Thus we hypothesize that regularizers in the form of BC regularizer, or CQL-style regularizer in our case, may be needed in these domains due to the nature of these environments. While the answer to when such regularizers are needed is not yet clear, we view it as a strong contribution that our work engenders such questions on the nature of offline RL problems.
> > - Response to Major Questions:
> >   - 1) Throughout our experience in this project we have tried both forms of Shared-Targets, i.e. double-pessimism, as well as the method you suggested where we use the mean of the Q-functions in the policy update procedure. Experimentally we did not see a significant difference between the two approaches, and decided to present our results using double-pessimism as it is the closest to how Shared-Target ensembles are currently employed in the deep RL literature. Additionally, we would like to emphasize that Theorem 3.1 and Section 3.2 is not impacted by the choice of “double-pessimism” or not, as Theorem 3.1 and the toy construction in Section 3.2 are concerned with pessimistic policy evaluation, and not the policy optimization aspect of the algorithm.
> >   - 2a) Mathematically speaking, under the assumption of Theorem 3.1, in the limit of infinite ensemble size, taking the min/inf over the ensemble will result in $\forall (s,a), Q(s,a) \rightarrow -\infty$. For this reason, we did not build the MSG algorithm off of the min formulation, but instead used the LCB formulation. We did however perform an experiment on the D4RL antmaze medium-play, medium-diverse, large-play, and large-diverse, using the Independent min formulation and ensemble size 64, with the same hyperparameter search range as in Figure 3 for the $\alpha$ parameter (note that in the min formulation there is no longer a $\beta$ parameter). We observe that on the antmaze-medium domains MSG-Min may be more robust, while on the antmaze-large domain, MSG-LCB may lead to better results. We have included our results in a new Appendix K in the updated rebuttal revision pdf.
> >   - 2b) This is an interesting suggestion, however we have not experimented with this approach. Would you be able to elaborate on your intuition for this approach?

---

> > > ### Author Response · Authors · 2022-08-04
> > > **Response to Reviewer nkxP (Part 3)**
> > >
> > > - Response to Minor Questions:
> > >   - $\gamma$ in Section 3.2: We experimented with a couple of values for $\gamma$ and used one that had a more robust effect. Using smaller $\gamma$ also helps $\gamma \vert\vert C \vert\vert < 1$. In general, regardless of the ensemble setting and the topic of our work, when $\gamma \vert\vert C \vert\vert \geq 1$, the policy evaluation process is liable to diverge.
> > >   - $y$-axis in Figure 2 and 3: Thank you, indeed Figure 3’s $y$-axis should be multiplied by 100.
> > >   - Please refer to earlier above for our discussion on Figure 3. But as a direct answer to your question, each dot represents the final policy performance for one experiment (i.e. one specific combination of domain, hyperparameters, and random seed). The colors, as noted in the caption of Figure 3, represent the value of the $\alpha$ hyperparameter.
> > >   - Line 763: Thank you for your detailed review of our work! Indeed $\theta^{(0)}_{\text{lin}} = \theta^{(0)}$. This has been updated in our rebuttal revision pdf.
> > > - Response to Limitations:
> > >   - 1) Please refer to our detailed response to your comment for Weakness #2.
> > >   - 2) Thank you for your acknowledgement of the strengths of our work! We hope that the main thread of our work, in addition to future research directions highlighted by experiments in Section 5.3 (Efficient Ensembles), become a valuable contribution to the offline RL literature.
> > >
> > > References:
> > > - [1] Towards Characterizing Divergence in Deep Q-Learning
> > > - [2] EMaQ: Expected-Max Q-Learning Operator for Simple Yet Effective Offline and Online RL
> > > - [3] Offline Reinforcement Learning with Implicit Q-Learning
> > > - [4] Decision Transformer: Reinforcement Learning via Sequence Modeling
> > > - [5] AWAC: Accelerating Online Reinforcement Learning with Offline Datasets
> > > - [6] Offline RL Without Off-Policy Evaluation
> > > - [7] A Minimalist Approach to Offline Reinforcement Learning

---

> > > > ### Comment · Reviewer_nkxP · 2022-08-07
> > > > **Responses and elaborations**
> > > >
> > > > I appreciate the authors' detailed responses and more updates in the manuscripts according to the reviews' comments.
> > > >
> > > > 1. First I appreciate the additional experiments made during the rebuttal period that sheds light on the empirical difference between using min of the ensemble vs. using lcb. The result is interesting. However, I would respectfully disagree with the authors on the reason not to use min of the ensemble: first we should reasonably assume that every function in our function class Q should be reasonably bounded (for example, $[0, V_{max}]$). Also, we should always assume or enforce some nice properties of our function class (for example, Bellman completeness as in [1]), so that even taking min of the entire function class (not just the ensemble!) should still give us something reasonable. Still, the nuances in practice would still be an interesting problem to investigate and again I appreciate the authors' efforts to perform this additional evaluation that may just be of independent interest.
> > > >
> > > > 2. Regarding different hyperparameters reported for the evaluations, there seems no issue now if every baselines are tuned and reported in the same way.
> > > >
> > > > 3. To elaborate my Q2(b): what I meant in the original review is: for each q in our ensemble class, we perform update in the shared target mannor (or instead of \EE[Q] - std[Q], we set the target as Q_i - std[Q]), and we just use \EE[Q] for policy optimization. In this way we only perform pessimism once (during the value update), but I admit this seems a less intuitive way than the proposed method.
> > > >
> > > >
> > > > [1] Bellman-consistent pessimism for offline reinforcement learning.

---

> > > > > ### Author Response · Authors · 2022-08-09
> > > > > **Thank you for your engagement in our review process!**
> > > > >
> > > > > - Q1) We completely agree we your statement. What we wanted point at is that in the NTK setting of Theorem 3.1, the outputs of the value functions are normally distributed. Since we did not make additional assumptions, such as limiting the bounds of the function class, an $\inf$ formulation would have resulted in $-\infty$. Using an LCB also had the additional advantage of providing us with the $\beta$ hyperparameter, which can be used as a knob for tuning the extent of pessimism. This knob proved to be useful in benchmark experiments, such those in Figure 3 and Appendix K.
> > > > > - Q3) This definitely sounds like a reasonable suggestion, since each individual ensemble member's targets contain independent as well as shared components.
> > > > >
> > > > > We greatly appreciate your detailed feedback throughout the review process!

---

### Official Review · Reviewer_JRTT · 2022-07-09

**Rating:** 6
**Confidence:** 3
**Soundness:** 3 good
**Presentation:** 3 good
**Contribution:** 3 good

**Summary:**

This paper discusses the uncertainty estimation in RL, which is an alternative to induce pessimism in offline RL. Though uncertainty estimation through ensembles has been proposed in the offline RL literature, this paper points out a critical flaw in how to incorporate the LCB in the actor-critic based algorithm. It shows theoretically that the previous algorithms, which regress different Q functions to the shared  pessimistic target values, and does policy evaluation based on the LCB could sometime leads to over-estimation of the Q function. To address this, the paper proposes a simple fix, that is in the Bellman backup stage, instead of regressing the different Q values to the shared LCB estimate, just regressing them to independent Q target. Empirically, it shows better performance in challenging tasks that require stitching. Beyond this, the paper also examines how different efficient ensemble method work in RL setting, and it seems there still exist a large gap in the performance compared with deep ensemble, which opens up more interesting questions in efficient ensemble methods in the RL setting.

**Questions:**

[1]. The comparison of independent and shared target comparison is interesting. Could the authors summarize the literature in a clearer way in how they do ensembles, and mark the subtle difference, as this is missing in Section 2, but these subtle differences seem make a big difference, as what is discussed in this paper.
[2]. From L154-155, it seems the shared targets are doing this "doubling pessimism" in both the policy evaluation and learning step, which seems actually a little weird to me, could the authors point out the exact literature that is doing this. As I can imagine either (1). we are doing the independent target, or (2). we do the evaluation pessimistically, but in learning we take the mean of the Q, compared to LCB. Just curious that do the authors try (2)?
[3]. Followed by [2], it seems the shared target do seem doing more pessimism, but theoretically it could lead to over-estimation. Any comments in bringing this intuition to the theoretical results? it seems related with the correlation of the transition (s,a) paris and properties of the C matrix. Could the authors comment more on when the shared target leads to over-pessimism or optimism, and the practical guidelines?
[4]. The toy example in Section 3.2 is a little bit confusing to me, what is the rationale of using the policy that play the same a in the next state?
[5].The discussion in L327-337 shows that the shared target has worse performance as ensemble sizes decreases, any explanation on this?

**Ethics Review Area:**

["I don’t know"]

**Limitations:**

Yes.

**Strengths And Weaknesses:**

Strength:
[1]. The paper is well-written and very easy to follow.
[2]. It discusses a major flaw in how to incorporate LCB estimate in offline RL algorithms, which seems being overlooked in the literature, but empirically seems make a big difference. The theoretical claim is well-supported.
[3]. It has a comprehensive set of empirical studies, which covers various aspects about the applicability of the method, such as the ensemble size, the hyper-parameter sensitivity. I do appreciate the authors' effort in discussing how to transfer efficient ensemble method in supervised learning setting to the RL setup, to make it more computationally efficient, though some negative results there.

Weakness:
[1]. Maybe I am misunderstanding something, but i do feel some of the claims and findings are not explained in a super clear way, see Questions section for details for this.
[2]. The experiment section does show that incorporating the independent target leads to the better performance in the challenging tasks, it would be great to see that this is result from better LCB estimate. We see the overestimation issues in the toy task, and it would be really helpful to see in the challenging tasks, that shared target does lead to over-estimation, which is the reason that the method helps.
[3]. Section 4.2 seems a little bit out of picture. As it seems alpha=0 works great in most cases, the authors state that it might help in some narrow data regime, is any empirical study supporting this?

---

> ### Author Response · Authors · 2022-08-04
> **Response to Reviewer JRTT (Part 1)**
>
> Thank you for your detailed review of our work! We hope that our response below addresses your concerns. We look forward to a productive Author-Reviewer discussion period.
>
> - Regarding Strengths:
>   - We appreciate your acknowledgement of our results concerning efficient ensembles, and the future directions they indicate.
> - Regarding Weaknesses:
>   - 2) Analyzing improved LCB estimates:
>     - We were wondering what form of experiment you were thinking could help analyze the quality of LCB estimates in the benchmark domains? Did you have in mind something along the lines of fixing some policy, and performing pessimistic policy evaluation under both settings (shared and independent targets), and comparing the LCB estimates?
>     - While we did not have room to include Appendix G in the main manuscript, we believe that the toy (but close to practical) pedagogical experiments in this section also provide important intuitions regarding the behavior of ensembles under the various settings. We believe this might be of interest to you, given your question.
>     - Given the intuitions garnered from Theorem 3.1 + Section 3.2 + Appendix G, we believe that the problem of Shared-Target ensembles goes beyond the optimistic LCB problem: Theorem 3.1 + Section 3.2 showed that optimistic estimates can and do happen, thus the Shared-Targets formulation is mathematically flawed. However, we believe the Shared-Targets formulation is also “philosophically” flawed as well. Following the discussion in Lines 186-190, when we train a $Q$-ensemble independently, in a sense we are estimating a belief distribution over $Q$-functions, and in the policy optimization step we optimize with respect to an LCB on this distribution. In Shared-Target ensembles however, there is no such interpretation, and all the $Q$-functions rapidly collapse to being very similar (mathematically this collapse is also seen in equation 32). These intuitions are also verified in Appendix G.1. In addition, there are effects that arise when we move from infinite-width NTK settings towards the practical finite-width setting. These effects which we do not yet understand are empirically demonstrated in Appendix G.2.
>     - In the practical deep RL setting, other factors such as not being in the infinite-width NTK setting, the use of target Q-networks, not performing policy evaluation until close to convergence, using Adam optimizer, etc. also come into play which makes rigorous analysis very difficult.
>   - 3) Regarding section 4.2:
>     - We agree with you that even with $\alpha=0$ we obtain very strong results. Figure 3 presents the results for 5 random seeds, per ($\alpha$, $\beta$, ensemble size) hyperparameter combination, with each dot representing one experiment. In Figure 3, the following can be clearly observed. 1) As the ensemble size increases, results improve, in particular for the two hardest antmaze-large domains, showing the significant impact of offline RL through ensembles, as larger ensembles are closer to the theoretical setting. 2) Looking at ensemble size 64, for the two antmaze-large domains almost all of the blue dots ($\alpha=0$, no CQL-style regularizer) are obtaining SOTA results, and for the two antmaze-medium domains, we see that by increasing the amount of pessimism only through the LCB (i.e. setting $\beta=-8$), the blue dots ($\alpha=0$, no CQL-style regularizer) are again amongst the best results we obtain. This again, supports the statement that uncertainty estimation through ensembles can indeed be used as the main, and only, source of pessimism for offline RL. 3) The ablation results in Figure 13 (in the appendix) demonstrate that if we modify the implementation of ensembles in 1 line of code, and go from Independent Ensembles to Shared-LCB or Shared-Min Ensembles, the SOTA results we obtained become 0 across the board, for every domain, hyperparameter, and random seed. This ablation undeniably demonstrates the fundamental practical significance of Theorem 3.1 and Section 3.2.
>     - The main domains where we have observed the need for $\alpha > 0$ is in the D4RL gym domains (HalfCheetah, Hopper, Walker2d). In prior work such as EMaQ [6], it has been shown that performant policies for these domains can be surprisingly close to a behavioral cloning policy. Additionally, in IQL [5], it has been shown that prior methods such as 10% BC, Decision Transformers [8], AWAC [9], Onestep RL [10], TD3+BC [7], which are very close to behavioral cloning, do very well on these gym domains, while completely failing on on the antmaze domains. Thus we hypothesize that regularizers in the form of BC regularizer, or CQL-style regularizer in our case, may be needed in these domains due to the nature of these environments. While the answer to when such regularizers are needed is not yet clear, we view it as a strong contribution that our work engenders such questions on the nature of offline RL problems.

---

> > ### Author Response · Authors · 2022-08-04
> > **Response to Reviewer JRTT (Part 2)**
> >
> > - Response to Questions:
> >   - 1) The use of Shared-Target ensembles has become an almost default implementation choice in all actor-critic methods, whether offline or online, since its introduction in the work of [1] (section 4.2, Clipped Double Q-Learning for Actor-Critic). As discussed in Section 3.1, this approach advocates for computing TD targets as $y^i = r(s,a) + \gamma min_{i}[Q^i(s’, \pi(s’))]$. In actor-critic offline RL (which is the most popular subset of deep offline RL methods), Shared-Target ensembles are universally used as well. Examples include popular algorithms such as: BCQ [2], BEAR [3], CQL [4], IQL [5], EMaQ [6], TD3+BC [7].
> >     - Note that in our work we use LCB instead of the min for our analysis, and report empirical results for both Shared-LCB and Shared-Min baseline. We also tried an experiment with MSG using min instead of LCB, which is now available in Appendix K of our updated rebuttal manuscript.
> >   - 2) Throughout our experience in this project we have tried both forms of Shared-Targets, i.e. double-pessimism, as well as the method you suggested where we use the mean of the Q-functions in the policy update procedure. Experimentally we did not see a significant difference between the two approaches, and decided to present our results using double-pessimism as it is the closest to how Shared-Target ensembles are currently employed in the deep RL literature. Additionally, we would like to emphasize that Theorem 3.1 and Section 3.2 is not impacted by the choice of “double-pessimism” or not, as Theorem 3.1 and the toy construction in Section 3.2 are concerned with pessimistic policy evaluation, and not the policy optimization aspect of the algorithm.
> >   - 3) Indeed, as you noted, whether overestimation happens or not is due to a very complex interaction between the C matrix, the neural network architecture, the initial weight distribution used to initialize the weights of the networks in the ensemble (since it affects the distributions of $Q^{(0)}(s,a)$), the data available in the offline dataset, and the policy being evaluated. The C matrix is itself a direct function of the offline dataset + the neural network architecture + the policy being evaluated: $C := \hat{\Theta}^{(0)}(X’, X) \cdot \hat{\Theta}^{(0)}(X, X)$, where $X’ := \text{data matrix }(s’, \pi(s’))$, $X := \text{data matrix }(s,a)$, and $\hat{\Theta}$ is the NTK kernel corresponding to the specific neural network architecture used. Due to such complex interactions, it can be quite challenging to intuitively understand when the problem with optimism can occur. Looking at the equations, if the C matrix only contains non-negative entries then it would avoid the optimism problem with Shared-Targets (however, it will not resolve the problem of “correctness” of Shared-Targets, in the sense that they do not correspond to a notion of LCB on a posterior of Q-functions). A special case would be if the C matrix was a diagonal matrix with non-negative entries. A special case where this would happen is if, for all $(s,a)$, $(s,a)$ is only strongly correlated with itself and $(s’, \pi(s’)$, with the notion of correlation coming from the NTK, i.e. $\hat{\Theta}^{(0)}(X, X)$ and $\hat{\Theta}^{(0)}(X’, X)$.
> >   - 4) In designing the toy example in Section 3.2 our goal was to design a simple toy domain. The sole reason for the choices we made (such as normal distributions, and choice of policy) was to keep our example as simple and straightforward and easy to describe as possible. Appendix G also includes additional toy experiments providing intuitions on Independent vs. Shared.
> >   - 5) As discussed earlier above in this response, we believe that the optimism issue that we highlight in this work points at a broader problem, demonstrating that Shared-Targets can be fundamentally problematic, and not the correct way to train ensembles of Q-functions. In our intuition, idiosyncrasies such as the one you highlighted (Line 327-337 referencing Figure 7) further support that Shared-Targets are a problematic formulation.
> >
> > References:
> > - [1] Addressing Function Approximation Error in Actor-Critic Methods
> > - [2] Off-Policy Deep Reinforcement Learning without Exploration
> > - [3] Stabilizing Off-Policy Q-Learning via Bootstrapping Error Reduction
> > - [4] Conservative Q-Learning
> > - [5] Offline Reinforcement Learning with Implicit Q-Learning
> > - [6] EMaQ: Expected-Max Q-Learning Operator for Simple Yet Effective Offline and Online RL
> > - [7] A Minimalist Approach to Offline Reinforcement Learning
> > - [8] Decision Transformer: Reinforcement Learning via Sequence Modeling
> > - [9] AWAC: Accelerating Online Reinforcement Learning with Offline Datasets
> > - [10] Offline RL Without Off-Policy Evaluation

---

### Official Review · Reviewer_nPdV · 2022-07-11

**Rating:** 5
**Confidence:** 4
**Soundness:** 3 good
**Presentation:** 3 good
**Contribution:** 2 fair

**Summary:**

This paper studies ensemble-based pessimism in offline RL from both theoretical and empirical aspects,
- Giving an analysis mathematically through NTK to show shared pessimistic targets can paradoxically lead to Q-estimates which are in fact optimistic.
- Proposing MSG that trains each Q-network independently, and conducts experiments in D4RL and RL Unplugged tasks.



**Questions:**

- In line 191, why the pessimism term of method 2 may become positive, while it cannot be positive in method 1?
- ln line 781, how to get Eq. 20, 21, 22 from Eq. 19?
- ln line 791, how to get Eq. 31, 32 from Eq. 30?


**Ethics Review Area:**

["I don’t know"]

**Strengths And Weaknesses:**

Strengths
- Formally analyze the offline RL methods based on Q-ensemble pessimism on infinite-width neural networks, and shows the pessimism term may become positive in shared targets.
- Empirically verifies the effectiveness of the algorithm by combining Q-ensemble with CQL.

Weaknesses
- The NTK assumptions in section 3.1 and the Gaussian assumptions in section 3.2 seems limited in broader value iteration.
- The Q-ensemble needs to combine with CQL to obtain reasonable performance. The use of CQL makes it difficult to analyze the source of performance improvement. However, there exist several methods (e.g., EDAC in NeurIPS 2021 and PBRL in ICLR 2022) that perform Q-ensemble for offline-RL with purely uncertainty.
- The experiment results are not complete for D4RL benchmark.

---

> ### Author Response · Authors · 2022-08-04
> **Response to Reviewer nPdV (Part 1)**
>
> Thank you for your detailed review of our work! We hope that our response below addresses your concerns. We look forward to a productive Author-Reviewer discussion period.
>
> - Regarding Strengths:
>   - We would additionally like to emphasize the following:
>     - The magnitude of performance gains on the very challenging D4RL antmaze domains is quite significant (as compared to current SoTA IQL [1] in Table 1).
>     - In addition to the main thread of our work, we included extensive results with various state-of-the-art efficient ensemble approaches from the supervised learning literature. Our experiments demonstrated that they cannot match the performance of MSG with full separate ensemble networks, which we believe is an important result that highlights valuable avenues of future research in offline RL.
>
> - Regarding Weaknesses:
>   - NTK assumption:
>     - In the current deep learning literature, the NTK assumption is one of the main tools used in the literature to understand RL algorithms using deep neural networks. Many works, such as [2], have used the NTK setting to study questions related to deep RL, with findings being relevant to the practical settings. Additionally, the results of our ablations clearly demonstrate that resolving the problem discussed in Theorem 3.1 and Section 3.2 is the key enabler of successful ensemble-based pessimism for offline RL (details below).
>   - Gaussian assumption:
>     - We would like to emphasize that the Gaussian distributions are not related to the proof of Theorem 3.1, and are only used the construction of the toy MDP and dataset in Section 3.2. In Theorem 3.1 we use the NTK assumption to compute closed form equations for the predictions of the Q-ensembles under Independent vs. Shared Targets Q-ensembles. Equation 2 in Theorem 3.1 suggests that over-estimation *could potentially happen*. In section 3.2, we construct a toy offline RL setup, and demonstrate that in the non-NTK setting, with finite datasets and network widths, over-estimation *does happen*. The Gaussian distributions are solely used in the construction of this toy MDP.
>     - Prior offline RL methods using uncertainty estimation techniques:
>       - EDAC [2] uses Shared-Min Targets for training offline RL policies. However, they observe that they sometimes require excessively large ensemble sizes (N=500) for stable learning. Thus, they introduce an additional loss function to diversify Q-functions in the ensemble. Indeed, our results shed light on why this diversification loss is necessary. In our proof of Theorem 3.1 (Appendix F), we can see that using shared targets results in a collapse of variance in the Q-functions (equation 32). This collapse is also evidences by experiments in a pedagogical toy MDP in Appendix G, Figure 17, where we observe that whenever targets are shared, there is a significant collapse of uncertainty. As a note, the experiments in Appendix G are separate from Section 3.2 and we believe they provide intriguing intuitions regarding independent vs. shared target ensembles.
>       - PBRL [4], motivated by [5], take a different approach for LCB-based policy evaluation. They make use of two objectives for training their policies: 1) While they do not discuss any matters relating to Theorem 3.1 and Section 2, they use a mix of shared and independent targets for pessimistic policy evaluation. 2) A regularization objective that pushes down on the Q-values of actions not seen in the dataset.

---

> > ### Author Response · Authors · 2022-08-04
> > **Response to Review nPdV (Part 2)**
> >
> > - Regarding Weaknesses:
> >   - Combining with CQL-style regularizer:
> >     - Figure 3 presents the results for 5 random seeds, per ($\alpha$, $\beta$, ensemble size) hyperparameter combination, with each dot representing one experiment. In Figure 3, the following can be clearly observed. 1) As the ensemble size increases, results improve, in particular for the two hardest antmaze-large domains, showing the significant impact of offline RL through ensembles, as larger ensembles are closer to the theoretical setting. 2) Looking at ensemble size 64, for the two antmaze-large domains almost all of the blue dots ($\alpha=0$, no CQL-style regularizer) are obtaining SOTA results, and for the two antmaze-medium domains, we see that by increasing the amount of pessimism only through the LCB (i.e. setting $\beta=-8$), the blue dots ($\alpha=0$, no CQL-style regularizer) are again amongst the best results we obtain. This again, supports the statement that uncertainty estimation through ensembles can indeed be used as the main, and only, source of pessimism for offline RL. 3) The ablation results in Figure 13 (in the appendix) demonstrate that if we modify the implementation of ensembles in 1 line of code, and go from Independent Ensembles to Shared-LCB or Shared-Min Ensembles, the SOTA results we obtained become 0 across the board, for every domain, hyperparameter, and random seed. This ablation undeniably demonstrates the fundamental practical significance of Theorem 3.1 and Section 3.2.
> >     - The main domains where we have observed the need for $\alpha > 0$ is in the D4RL gym domains (HalfCheetah, Hopper, Walker2d). In prior work such as EMaQ [10], it has been shown that performant policies for these domains can be surprisingly close to a behavioral cloning policy. Additionally, in IQL [1], it has been shown that prior methods such as 10% BC, Decision Transformers [7], AWAC [8], Onestep RL [9], TD3+BC [11], which are very close to behavioral cloning, do very well on these gym domains, while completely failing on on the antmaze domains. Thus we hypothesize that regularizers in the form of BC regularizer, or CQL-style regularizer in our case, may be needed in these domains due to the nature of these environments. While the answer to when such regularizers are needed is not yet clear, we view it as a strong contribution that our work engenders such questions on the nature of offline RL problems.
> >     - More D4RL results:
> >       - From the Gym domains (HalfCheetah, Hopper, Walker2d), the only setting we did not report results on was the “random” data setting.
> >       - We have reported results on all D4RL antmaze settings.
> >       - We only did not run experimentation on the 3 kitchen domains, and the Adroit domains. We did not run experiments on the D4RL Adroit domains because in communication with the creators, we have identified important problems in the way the Adroit domains are set up. Specifically, the eval horizons are very significantly shorter than the number of timesteps the trajectories in the datasets take to succeed. As can be seen by all works that report results on Adroit, the only domain for which they can report non-trivial results are the “pen” domains, or “expert” domains which fall under imitation learning rather than offline RL.
> >       - Nonetheless, to provide more results, we presented experiments on the RL Unplugged benchmark as well.
> >
> > - Response to Questions:
> >   - Line 191: When we compare equations 1 and 2 in Theorem 3.1, we observe that the first two terms are identical, and the only difference is in the third term, which corresponds to the pessimism term. As a reminder, note that in our notation, the square-root operation is applied element-wise to each entry in the corresponding vector. When using Independent Targets, we have the negative of a square-root term, which will always be a negative value. However, when using Shared Targets, since the “backup term” is being multiplied by the square root term, depending on the $C$ matrix and the values of $Q^{(0)}_{\theta^i}$, the resulting values can be positive or negative.
> >   - Line 781: As a reminder, expectations and variances are computed with respect to the initial weight distribution of the neural networks, i.e. our question is asking if we sample random initial weights for the Q-function from the initial weight distribution, what is the expected value and variance of the Q-values and $t$ iterations of policy evaluation. Also as a reminder, in our notation, expectation and variance are computed element-wise for each entry in the vector. Under the NTK setting, according to [12], $\forall (s,a), E_{ens}[Q^{(0)}(s,a)] = 0$. This immediately provides equation 20 from equation 19. Using equation 20, we can obtain the equation for the Variance (equations 21, 22) using the variance formula: $V_{ens}(X) = E_{ens}[(X - E_{ens}[X])^2]$.

---

> > > ### Author Response · Authors · 2022-08-04
> > > **Response to Review nPdV (Part 3)**
> > >
> > > - Response to Questions:
> > >   - Line 791: With the same reasoning, from equation 30 we obtain the expectation equation 31, and plug into the variance equation to obtain equation 32.
> > >
> > > References:
> > > - [1] Offline Reinforcement Learning with Implicit Q-Learning
> > > - [2] Towards Characterizing Divergence in Deep Q-Learning
> > > - [3] Uncertainty-Based Offline Reinforcement Learning with Diversified Q-Ensemble
> > > - [4] Pessimistic Bootstrapping For Uncertainty Driven Offline Reinforcement Learning
> > > - [5] Is Pessimism Provably Efficient for Offline RL?
> > > - [6] Offline Reinforcement Learning with Implicit Q-Learning
> > > - [7] Decision Transformer: Reinforcement Learning via Sequence Modeling
> > > - [8] AWAC: Accelerating Online Reinforcement Learning with Offline Datasets
> > > - [9] Offline RL Without Off-Policy Evaluation
> > > - [10] EMaQ: Expected-Max Q-Learning Operator for Simple Yet Effective Offline and Online RL
> > > - [11] A Minimalist Approach to Offline Reinforcement Learning
> > > - [12] Wide neural networks of any depth evolve as linear models under gradient descent

---

### Official Review · Reviewer_qbbN · 2022-07-12

**Rating:** 4
**Confidence:** 4
**Soundness:** 2 fair
**Presentation:** 3 good
**Contribution:** 2 fair

**Summary:**

The paper revisits the design of ensemble critics in offline RL. The authors argue that the common design where critics in the ensemble sharing the same, pessimistic target function in learning can lead to actually optimistic critics. The authors analyze this phenomenon theoretically under the NTK assumption, and present toy simulation examples. The authors further use this insight to design an offline RL algorithm MSG, which gives SoTA results on common offline RL benchmarks. In these experiments, they show that the separating the targets is a key to the algorithm's superior performance.



********** Comments for Rebuttal *************


Thanks for the rebuttal and the clarification. While they address some of my concerns, my main concern stay the same. As stated in the original review, the main issue I have is "whether the proposal here is sufficient to design a full offline RL algorithm or just provide an important note on implementation choice". The rebuttal also points out "it is the objective of our work to advocate for relying on uncertainty estimation as the main source of pessimism for offline RL".

Let's examine this question from two aspects based on the paper and rebuttal .

From the theoretical side, the paper provides that Theorem 3.1, which compares the iterates of $Q_{LCB}$ of Independent Targets and the Shared Targets. However, it does not show "how"  good pessimistic the estimate of Independent Targets is. In the review, I mentioned "in general optimizing a pessimistic critic or being more pessimistic does not imply good performance in offline RL." because whether a pessimistic critic is useful or not depends on how "tight" the under estimation is and where it is pessimistic. Being more pessimistic is not always good, e.g., estimating all values as $V_{min}$ is pessimistic but it's obviously useless. The current theory does not provide enough insights to how good the pessimistic estimate is, or how good the learned policy based on such value estimate will be. This is why I said "the significance of the theoretical results are rather limited" in the review.

For empirical side, I think to demonstrate the authors claim, it is necessary to show that SoTA can be achieved with $\alpha=0$. However, the current results do not support that fully. While I agree that in Figure 3, $\alpha=0$ is among the best performing results. I also think that Figure 3 does not provide a conclusive answer, as it is missing results of larger $\alpha$ value for $\beta=0$, as there is an increasing trend. This was pointed out in the review. It's also hard to compare Fig 3 and Fig 2 directly

I think the failure of using $\alpha=0$ in simpler mujoco domains is actually showing that the proposed approach "alone" might not be sufficient to provide enough pessimism "broadly". I do not agree with the rebuttal's statement that these environments can be solved by behavior cloning. All the methods the authors listed there are not pure behavior cloning, which mimics all actions, as all of them perform some reasoning what actions are better based on the rewards. Therefore I don't think this is a good excuse of using the CQL term here. Yes, I agree that the proposed method works with $\alpha=0$ with Antmaze which is considered as a harder domain, but is it because of the reason that the authors mentioned? if this is the case, why does not perform well in simpler domain. Or is it because of something that is related to the structure of Antmaze environment and dataset? Currently, we don't have sufficient evidence to tell.

Thus, I think that currently the paper provides insufficient results to show that the proposed uncertainty estimation "alone" can achieve SoTA offline RL results. Nonetheless, I also think that this paper provides more than sufficient reasons to show that it would be a good design choice to improve an existing value-based offline RL algorithm. Therefore, I keep my original recommendation.



**Questions:**

My main concern is whether the independent ensemble idea here is indeed sufficient to induce adequate pessimism for offline RL as the author claim. Here're few reasons why it does not seem to be the case.

1.  Theorem 3.1 compare the results of a pessimistic policy evaluation procedure. The authors argue that using Shared Targets may lead to optimism, compared with the Independent Targets version. However, significance of such a comparison is quite limited for the following reasons. First, to my knowledge, I do not know of existing offline RL methods introducing pessimism purely based on what the authors describe in the Shared Targets update rule. While that update rule are used in some implementations, that is usually not the main source of pessimism that the offline RL algorithm uses but just a detail. So the theoretical comparison here may not be realistic, even if we acknowledge the NTK approximation part. Second, the authors study a version based on E[Q] - sqrt{V[Q]}. A more common implementation I know is pointwise pessimism, i.e. min_i Q_i(s,a), which e.g. is used in CQL and TD3+BC implementation based on double Q networks. I doubt the comparison would carry through to this more realistic case, because using pointwise pessimism would be strictly more pessimistic than the Independent Targets.

2. The above theoretical analysis is limited to offline policy evaluation. It's unclear how it can be translated into policy optimization, as in general optimizing a pessimistic critic or being more pessimistic does not imply good performance in offline RL. Therefore, the significance of the theoretical results are rather limited in my mind, unless the authors discuss in details how the proposed procedure affect the performance of the learned policy.

3. Another reason why the pessimism induced by Independent Training is not sufficient is that in MSG, the authors introduce the difference term H(\thteta^i) as part of the objective. The authors write "Empirically, as evidenced by our results in Appendix A.2, we have observed that such a regularizer can be necessary" This seems to imply that  the pessimism of the Independent Target can be serve as the main source of pessimism. In particular, the authors suggests using this extra regularizer H(\theta^i) is necessary, when the data distribution is narrow. Since addressing the lack of coverage is the main focus of offline RL (say compared with off-policy RL), this shows again the proposed pessimism is not sufficient for offline RL.

4. The above insufficiency is partially shown in the experimental results as well. For instance in Figure 3, MSG does not perform the best when alpha=0. In the experiments of beta=0, the authors should also try even larger alpha to get a fuller picture of the performance and how hyperparamters interact. From these results, it is unclear whether the good performance of MSG is due to using the ensemble uncertainty as the primary source of uncertainty.

In summary, while this paper presents an interesting observation, showing that using Independent Targets can be more pessimistic than using Shared Targets, such results do not directly address the main research question of offline RL for the reasons above. (But the title "Why So Pessimistic?" seems to want to convey that we should be less pessimistic?) It is therefore hard for me to position this paper. This paper presents a good and thorough empirical study on how a certain design knob in implementation can affect the performance. However, the results here do not provide enough insights to design offline RL methods more broadly outside the tested implementation here.




**Limitations:**

The authors well discuss the limitation such as the extra complexity needed by the proposed method. However, in my view, the results here are more limited than what the authors claim for the reasons above.

**Strengths And Weaknesses:**

Strengths:
1. This paper presents an overlooked finding.
2. It provides some theoretical reasons to back it up. It also provides empirical validation.
2. The writing of this paper is clear and the experimental results are thorough.

Weakness:
1. While the authors provide explanations of why the common usage of Shared Targets may lead to optimism, the current results are not conclusive. In particular, while existing offline RL implementations use Shared Targets, the pessimism of Shared Targets is not the main source of pessimism but rather an implementation detail. Therefore, it is unclear whether the proposal here is sufficient to design a full offline RL algorithm or just provide an important note on implementation choice. This factor limits the significance of the paper.

2. It is unclear what data assumptions are needed for the proposed method to work properly as an offline RL method.

---

> ### Author Response · Authors · 2022-08-04
> **Response to Reviewer qbbN (Part 1)**
>
> Thank you for your detailed review of our work! We hope that our response below addresses your concerns. We look forward to a productive Author-Reviewer discussion period.
>
> - Regarding Strengths:
>   - We would additionally like to emphasize the following:
>     - The magnitude of performance gains on the very challenging D4RL antmaze domains is quite significant (as compared to current SoTA IQL [4] in Table 1).
>     - In addition to the main thread of our work, we included extensive results with various state-of-the-art efficient ensemble approaches from the supervised learning literature. Our experiments demonstrated that they cannot match the performance of MSG with full separate ensemble networks, which we believe is an important result that highlights valuable avenues of future research in offline RL.
>
> - Regarding Weaknesses:
>   - Response to: “the pessimism of Shared Targets is not the main source of pessimism but rather an implementation detail … limits the significance of the paper.” :
>     - Most model-free algorithms in the deep offline RL literature rely on either 1) regularizing policies using behavioral cloning style objectives (e.g. TD3+BC [5], BCQ [7], BEAR [8], IQL [4]), or 2) Optimizing for pessimistic value function by pushing down on the values of out-of-dataset actions (e.g. CQL [6]). The purpose of our work is indeed **to be an advocate** for using uncertainty estimation for obtaining pessimistic value functions when designing an offline RL algorithm. While the approach of uncertainty estimation is well-motivated (Brockman et al. [1]), currently there is a  limited number of works attempting this direction, likely due to the challenging nature of uncertainty estimation, in particular when using neural network models. In supervised learning, ensembles are the preeminent approach to uncertainty estimation with deep models. Thus in our work, we studied if we can design offline RL algorithms that rely on uncertainty estimation through ensembles. On the path towards designing such an offline RL method, we discovered a key problem with how ensembles of value functions are typically trained (Theorem 3.1 and Section 3.2). Through extensive experiments and ablations, we demonstrated that resolving this problem is the key enabler of ensemble-based offline RL, and that we can obtain incredible results in very challenging offline RL benchmark tasks (further details below in response to another point). Thus the significance of our work is that: 1) We presented a successful offline RL algorithm that relies on uncertainty estimation techniques, 2) We discovered a mistake in a very common implementation choice across offline and online RL literature and demonstrated that this mistake is the key obstacle in making ensemble-based offline RL feasible (additional details below), 3) We obtained incredibly strong results on very challenging offline RL benchmarks.
>   - Data Assumptions:
>     - Discussed in response to Questions 3-4 below.
> - Response to Questions:
>   - 1a) “I do not know of existing offline RL methods introducing pessimism purely based on what the authors describe in the Shared Targets update rule … usually not the main source of pessimism that the offline RL algorithm uses but just a detail”: First, we would like to note that there are many works in offline RL where Shared Target ensembles are a key component of the devised method. As examples, BCQ [7], BEAR [8], EMaQ [3], TD3+BC [5], all operate by using a behavioral cloning regularizer/constraint plus Shared-Target ensembles for pessimism; also EDAC [2] solely relies on Shared-Target ensembles with an additional auxiliary loss to encourage greater diversity in the ensemble. Second, as discussed above, **it is the objective of our work** to advocate for relying on uncertainty estimation as the main source of pessimism for offline RL, which we accomplish through the use of ensembles. Our experiments demonstrate that resolving the problem discussed in Theorem 3.1 and Section 3.2 is the key enabler for obtaining very strong results on very challenging offline RL tasks (details in our response to your questions 3-4). We also believe that it is likely due to this problem that prior methods that try to rely on ensembles (e.g. EDAC [2]) required additional auxiliary objectives such as ensemble diversification losses.

---

> > ### Author Response · Authors · 2022-08-04
> > **Response to Reviewer qbbN (Part 2)**
> >
> > - Response to Questions:
> >   - 1b) In this work we are indeed talking about pointwise pessimism. The only difference is that instead of using the minimum operator in $min_i Q_i(s,a)$, we use the lower-confidence bound (LCB) operator $E_i [Q_i(s,a)] - \beta \cdot \sqrt{V_i[Q_i(s,a)]}$. There are two reasons for using the LCB. First, mathematically, under the NTK assumptions of Theorem 3.1, as ensemble size tends to infinity, for all $(s,a)$, $min_i Q_i(s,a)$ tends to negative infinity. Second, using LCB provides an additional knob $\beta$ for controlling the amount of pessimism. Using “Independent Targets” does not refer to pointwise pessimism. It refers to the two equations between lines 154-155, i.e. whether the target value $y^i$ is the same for all $i$, or each $Q_i$ computes $y^i$ based on its own value estimates.
> >   - 2) We believe your comment “as in general optimizing a pessimistic critic or being more pessimistic does not imply good performance in offline RL” is not an accurate statement. Devising correct approaches for obtaining pessimistic value estimates, and using them to optimize a policy is a well-established methodology in the offline RL literature (e.g. CQL [6], with more theoretical results in [1, 9, 10, 11]). The motivation of such approaches is that if we obtain reliable pessimistic value estimates, at test time the optimized policy is likely to obtain equal or higher returns than anticipated by the pessimistic values. Indeed, as an example, this is the approach taken by the very popular CQL algorithm, where the only source of pessimism is to push down on the values of actions not seen in the offline dataset.
> >   - 3-4) We would like to highlight that our experiments strongly support that **solely** using our proposed pessimism objective based on independent ensembles is sufficient for state-of-the-art (SOTA) offline RL performance (as a reminder, $\beta$ is the LCB hyperparameter, and $\alpha$ is the weight dictating the strength of the CQL-style regularizer):
> >     - Figure 3 presents the results for 5 random seeds, per ($\alpha$, $\beta$, ensemble size) hyperparameter combination, with each dot representing one experiment. In Figure 3, the following can be clearly observed. 1) As the ensemble size increases, results improve, in particular for the two hardest antmaze-large domains, showing the significant impact of offline RL through ensembles, as larger ensembles are closer to the theoretical setting. 2) Looking at ensemble size 64, for the two antmaze-large domains almost all of the blue dots ($\alpha=0$, no CQL-style regularizer) are obtaining SOTA results, and for the two antmaze-medium domains, we see that by increasing the amount of pessimism only through the LCB (i.e. setting $\beta=-8$), the blue dots ($\alpha=0$, no CQL-style regularizer) are again amongst the best results we obtain. This again, supports the statement that uncertainty estimation through ensembles can indeed be used as the main, and only, source of pessimism for offline RL. 3) The ablation results in Figure 13 (in the appendix) demonstrate that if we modify the implementation of ensembles in 1 line of code, and go from Independent Ensembles to Shared-LCB or Shared-Min Ensembles, the SOTA results we obtained become 0 across the board, for every domain, hyperparameter, and random seed. This ablation undeniably demonstrates the fundamental practical significance of Theorem 3.1 and Section 3.2.
> >     - The main domains where we have observed the need for $\alpha > 0$ is in the D4RL gym domains (HalfCheetah, Hopper, Walker2d). In prior work such as EMaQ [3], it has been shown that performant policies for these domains can be surprisingly close to a behavioral cloning policy. Additionally, in IQL [5], it has been shown that prior methods such as 10% BC, Decision Transformers [12], AWAC [13], Onestep RL [14], TD3+BC [5], which are very close to behavioral cloning, do very well on these gym domains, while completely failing on on the antmaze domains. Thus we hypothesize that regularizers in the form of BC regularizer, or CQL-style regularizer in our case, may be needed in these domains due to the nature of these environments. While the answer to when such regularizers are needed is not yet clear, we view it as a strong contribution that our work engenders such questions on the nature of offline RL problems.

---

> > > ### Author Response · Authors · 2022-08-04
> > > **Response to Reviewer qbbN (Part 3)**
> > >
> > > - Explaining the title:
> > >   - As discussed in our manuscript and earlier above in this response, most prior offline RL methods rely on some form of BC regularizer, or pushing down on the values of actions not seen in the offline dataset. In the theoretical discussion of Brockman et al. [1], it is suggested that such approaches correspond to a very pessimistic assumption that any action outside the dataset may be arbitrarily bad. Additionally, they discuss that estimating uncertainties of value functions would allow for a less pessimistic offline RL. In the current deep offline RL literature, there are very few methods that attempt to design an offline RL method through uncertainty estimation. In this work, we present MSG which leverages uncertainty estimation as the main form of pessimism for offline RL, and we see that it is very successful on the most challenging D4RL tasks, exceeding all prior results on these domains. Hence, we titled our work “Why so pessimistic?” to encourage researchers and practitioners to seriously consider uncertainty based methods for offline RL.
> > > - “However, the results here do not provide enough insights to design offline RL methods more broadly outside the tested implementation here.”:
> > >   - Our work demonstrates that uncertainty based methods can perform very well. Thus, this encourages future work to focus on improving uncertainty estimation techniques for offline RL, and studying the trade-offs between uncertainty based methods vs. behavioral cloning and other regularizers.
> > >   - Additionally, we demonstrate that efficient ensemble approaches from the supervised learning literature do not perform as well in the offline RL domain, encouraging future research into developing efficient uncertainty estimation techniques for offline RL.
> > >
> > > References:
> > > - [1] The Importance of Pessimism in Fixed-Dataset Policy Optimization
> > > - [2] Uncertainty-Based Offline Reinforcement Learning with Diversified Q-Ensemble
> > > - [3] EMaQ: Expected-Max Q-Learning Operator for Simple Yet Effective Offline and Online RL
> > > - [4] Offline Reinforcement Learning with Implicit Q-Learning
> > > - [5] A Minimalist Approach to Offline Reinforcement Learning
> > > - [6] Conservative Q-Learning
> > > - [7] Off-Policy Deep Reinforcement Learning without Exploration
> > > - [8] Stabilizing Off-Policy Q-Learning via Bootstrapping Error Reduction
> > > - [9] Provable Benefits of Actor-Critic Methods for Offline Reinforcement Learning
> > > - [10] Provably Efficient Offline Reinforcement Learning with Trajectory-Wise Reward
> > > - [11] Is Pessimism Provably Efficient for Offline RL?
> > > - [12] Decision Transformer: Reinforcement Learning via Sequence Modeling
> > > - [13] AWAC: Accelerating Online Reinforcement Learning with Offline Datasets
> > > - [14] Offline RL Without Off-Policy Evaluation

---

### Author Response · Authors · 2022-08-08
**Thank you for your detailed feedback; Authors-Reviewers discussion period coming to a close soon;**

We would like to thank all reviewers for their significant effort in providing us with high quality reviews with constructive feedback.

We hope that our responses have provided clarifications with respect to your comments. Our updated rebuttal pdf also contains the following modifications: Clarifying notations ($E_{ens}, V_{ens}$ in the equations), Visually improving Figures 1 and 2, and including Appendix Section K (comparing MSG with a min vs. LCB formulation) in response to a suggested experiment by Reviewer nkxP.

As the authors-reviewers discussion period is coming to a close soon, please do not hesitate to let us know if we can provide any further clarifications with respect to your comments.

Thanks you!

Paper7959 Authors

---

### Meta-Review · Area_Chair_7592 · 2022-08-22

**Recommendation:** Accept
**Confidence:** Less certain

**Metareview:**

The paper identifies a common flaw in pessimistic algorithms related to the use of shared targets, and propose an alternative based on independent targets that mitigate the overly-optimistic estimates. The rebuttal has addressed a number of concerns raised by the reviewers, and in particular, the negative reviewer qbbN acknowledged that

> ... the proposed idea here would make an existing algorithm that uses e.g. double Q networks (which is quite common) and also other main pessimism (like value penalty or closeness to behavior policy) to perform better. Thus, the insight here can be quite useful in practice.

That said, the reviewer is still concerned about the framing of the work

> the paper does not provide sufficient evidence (theoretically or empirically) that the proposed pessimistic estimate based on Independent Training "alone" is sufficient to design a SoTA offline RL algorithm [which the paper claims to]... I think that the paper needs to provide stronger evidences or changes the framing.

Given the strong support from other reviewers, the AC is leaning towards acceptance, but strongly recommend that the authors change the framing of the paper to honestly reflect the contributions of the work.

**Award:**

No

---

### Decision · Program_Chairs · 2022-09-14

Accept